# Exogenous Thymosin Beta 4 Suppresses IPF-Lung Cancer in Mice: Possibly Associated with Its Inhibitory Effect on the JAK2/STAT3 Signaling Pathway

**DOI:** 10.3390/ijms24043818

**Published:** 2023-02-14

**Authors:** Rui Yu, Dandi Gao, Jiali Bao, Ronghao Sun, Mengqi Cui, Yunyun Mao, Kai Li, Enbo Hu, Yanfang Zhai, Yanhong Liu, Yuemei Gao, Ting Xiao, Honggang Zhou, Cheng Yang, Junjie Xu

**Affiliations:** 1Institute of Biotechnology, Academy of Military Medical Sciences, Beijing 100071, China; 2State Key Laboratory of Medicinal Chemical Biology, College of Pharmacy and Tianjin Key Laboratory of Molecular Drug Research, Nankai University, Tianjin 300353, China

**Keywords:** IPF-LC, Tβ4, JAK2/STAT3

## Abstract

Idiopathic pulmonary fibrosis (IPF) is a chronic, progressive, fibrotic interstitial lung disease of unknown etiology. At present, the mortality rate of the deadly disease is still very high, while the existing treatments only delay the progression of the disease and improve the quality of life of patients. Lung cancer (LC) is the most fatal disease in the world. In recent years, IPF has been considered to be an independent risk factor for the development of LC. The incidence of lung cancer is increased in the patients with IPF and the mortality is also significantly increased in the patients inflicted with the two diseases. In this study, we evaluated an animal model of pulmonary fibrosis complicated with LC by implanting LC cells orthotopically into the lungs of mice several days after bleomycin induction of the pulmonary fibrosis in the same mice. In vivo studies with the model showed that exogenous recombinant human thymosin beta 4 (exo-rhTβ4) alleviated the impairment of lung function and severity of damage of the alveolar structure by the pulmonary fibrosis and inhibited the proliferation of LC tumor growth. In addition, in vitro studies showed that exo-rhTβ4 inhibited the proliferation and migration of A549 and Mlg cells. Furthermore, our results also showed that rhTβ4 could effectively inhibit the JAK2-STAT3 signaling pathway and this might exert an anti-IPF-LC effect. The establishment of the IPF-LC animal model will be helpful for the development of drugs for the treatment of IPF-LC. Exogenous rhTβ4 can be potentially used for the treatment of IPF and LC.

## 1. Introduction

Idiopathic pulmonary fibrosis (IPF) is a rare lung disease of unknown etiology. The median survival time after an IPF diagnosis is only 2–4 years [1]. At present, there are only two kinds of drugs to treat IPF, pirfenidone and nintedanib, which mainly delay the progression of the disease and improve the quality of life and the survival rate of patients [2]. Despite this, the mortality rate of IPF is still high, and the survival time is quite uneven.

Lung cancer (LC) is the most fatal disease in the world, and its morbidity and mortality are the highest. There are numerous pathogenic factors of LC. Recent epidemiological studies have shown that IPF is an independent risk factor for LC [3,4,5]. The risk of IPF patients developing lung cancer is eight times higher than that in the healthy population, with a prevalence from 2.7 to 48% and more than 50% in some cases [6,7,8]. In an epidemiological study in Republic of Korea, the prevalence of LC in IPF cases was 6.4%. Among patients without LC at the time of IPF diagnosis, the cumulative incidence of LC at 1, 3 and 5 years was 1.7%, 4.7% and 7.0%, respectively [9]. Thus, IPF increases the risk of LC.

There exists a close relationship between IPF and LC. Lung injury will lead to the activation of lung fibroblasts, abnormal accumulation of extracellular matrix, and the formation of pulmonary fibrosis. Cumulative genetic changes, activated mesenchyme cells and a continuously changing microenvironment promote the occurrence and development of LC [10]. Elderly male smokers with IPF and untreated patients with IPF have an even higher risk of LC. Some studies have found that the location of the LC in patients with IPF is in the areas affected by peripheral pulmonary fibrosis, especially in the lower lobe of the lung, which further supports the hypothesis that LC can be induced by IPF [11].

At present, there are no clear clinical guidelines for screening LC in IPF patients, and a large number of IPF patients with LC are diagnosed at their advanced stage manifested with progressed symptoms. There are no clear treatment guidelines for patients inflicted with IPF and LC. IPF’s treatment guidelines recommend that IPF be considered in the LC treatment decisions [12,13], but LC guidelines have not yet established a treatment plan for IPF-LC patients [13] and there is no consensus on the treatment of IPF-LC. With in-depth study of the pathogenic mechanism of the two diseases, an increasing number of studies have focused on the development of drugs with potential anticancer and antifibrotic effects, as well as the combined application of traditional antifibrotic and anticancer drugs. For example, the anti-fibrosis drug nintedanib has been approved as a second-line drug for the treatment of LC and has been shown to be effective in clinical studies. Some studies have shown that nintedanib may prolong the survival time of patients with IPF reduce the incidence of LC [14], but there is still a lack of studies on nintedanib in patients with IPF-LC. Although many studies are exploring the effect of anti-pulmonary fibrosis drugs in the treatment of LC and the role of anti-lung cancer drugs in the treatment of pulmonary fibrosis, they are limited to the treatment of a single disease. The treatment of LC with IPF should take the IPF component into consideration. Otherwise, acute aggravation of IPF and even health may occur with the treatment of the LC alone. Therefore, there is an urgent need to develop a usable animal model of IPF-LC for the evaluation of the efficacy and toxicity of drugs for the treatment of IPF-LC. In this study, we established an animal model of BLM-induced pulmonary fibrosis with in-situ lung cancer and evaluated the therapeutic effects of the thymosin β4 (Tβ4) as well as nintedanib and cyclophosphamide as the positive controls which have been proved to be effective against IPF and LC.

Tβ4 is a small molecular polypeptide of 43 amino acids with multiple biological functions, which is found in tissues, organs and cells of mammals and other vertebrates [15]. It has been reported that Tβ4 is involved in wound healing, inflammation, fibrosis and tissue regeneration [16]. However, while some studies have shown that Tβ4 can prevent bleomycin-induced pulmonary fibrosis, there are no reports on the effect of exogenous Tβ4 on lung cancer and IPF-LC. Considering that Tβ4 has been reported to have anti-inflammatory effects and inflammation is one of the mechanisms shared by lung cancer and pulmonary fibrosis, Tβ4 may have potential application value in the treatment of IPF-LC.

The recombinant human Tβ4 (rhTβ4) used in this study was prepared with our patented technology (Chinese invention patent No: ZL200910135972.0 and ZL201910498316.0). We also developed a method to acetylate serine residual at the N-acetylated of rhTβ4 in bacterial cells. The N-terminal serine residue of our rhTβ4 were 100% acetylated. Our rhTβ4 also showed excellent therapeutic effect on the rats with moderate to severe xerophthalmia [17], the ability to promote healing in a benzalkonium chloride (BAC)-induced mice dry eye disease (DED) model [18], improving scalp condition and microbiome homeostasis in seborrheic dermatitis [19] and increasing the survival rate of mice infected with MHV-A59 through inhibiting virus replication, balancing the host’s immune response, alleviating pathological damage, and promoting repair of the liver [20].

In this study, we used an established IPF-LC animal model to evaluate the anti-IPF-LC effect of rhTβ4, and exogenous rhTβ4 showed a good effect.

## 2. Results

### 2.1. Exogenous rhTβ4 Alleviates the Impairment of Lung Functions and Alveolar Strucure by IPF as Well as Growth of LC Tumor in Mice

The diagram of the experimental protocol with the mice is shown in Figure 1A. The body weight of the mice was recorded every day from the first day of modeling until sampling on the 21st day. The deceased mice were excluded, and the average weight of the mice in each group was calculated. Body weight changes in each group of mice are shown in Figure 1B. Nintedanib, cyclophosphamide and rhTβ4 improved the weight loss trend of the mice, especially in the rhTβ4 200 μg dose group where the weight increase trend was obvious.

The results of the measurement of the hydroxyproline content in the lung tissue of mice are shown in Figure 1C. The results showed that the content of hydroxyproline in the lung tissue of mice in the model group was significantly increased, and the positive control drugs nintedanib, cyclophosphamide and rhTβ4 significantly decreased it. The hydroxyproline content in the lungs of mice treated with rhTβ4 decreased in a concentration-dependent manner, and the hydroxyproline content in the rhTβ4 200 μg dose group was lower than that in the positive control drug group, indicating that this dose of rhTβ4 has a better anti-pulmonary fibrosis therapeutic effect.

The lung function test data of the mice are shown in Figure 1D. Compared with the model group, rhTβ4 treatment significantly improved the lung function of the mice, specifically forced vital capacity (FVC) and lung dynamic compliance (Cydn) were significantly increased, inspiratory resistance (Ri) and expiratory resistance (Re) were significantly decreased, and the rhTβ4 high-dose group had better effects than the positive control drug nintedanib and cyclophosphamide groups.

The HE and Masson staining results shown in Figure 1E,F show that, compared with the negative control group, the lung tissue structure of the mice in the BLM + LLC model group was severely damaged, the alveolar structure was incomplete, and fibrous foci appeared in the lungs; nintedanib and rhTβ4 treatment could significantly improve the pathological phenomenon of the mice. The degree of lung tissue damage and the number of fibrous foci was significantly reduced, and the pathological results showed that the treatment effect of rhTβ4 was slightly better than that of nintedanib.

The results of in vivo imaging showed that the tumor area of the mice in the BLM + LLC model group was larger, and the fluorescence value of the tumor fluorescence area was higher, while the positive control drugs nintedanib and cyclophosphamide and rhTβ4 could significantly reduce the tumor area and fluorescence value of the mice. Figure 2A shows that the positive control drugs and rhTβ4 have better antitumor effects, and the fluorescence values of the mice on the 7th, 14th, and 21st days were calculated, showing that the tumor fluorescence values of the mice increased sharply on the 14th to 21st days. Two positive drugs and rhTβ4 can be all slow tumor progression. Cyclophosphamide had the best effect on the tumor, while rhTβ4 could slow tumor progression in a concentration-dependent manner, as showed in Figure 2B.

### 2.2. Exogenous rhTβ4 Inhibits ECM Accumulation and Tumor Proliferation in IPF-LC Model Mice

The expression levels of fibrosis-related proteins were detected and are shown in Figure 3A–D. The expression levels of Col-1, fibronectin (Fn) and α-SMA in the model group were significantly increased, while rhTβ4 significantly reduced their expression. The expression of Col-1, Fn and α-SMA in lung tissue decreased proportionally to the concentration of rhTβ4, and rhTβ4 had a better effect than the other two drugs.

As shown in Figure 3E,F, immunohistochemical staining for the fibrosis markers Col-1, Fn and α-SMA and the tumor proliferation marker Ki-67 was performed on the lung tissue of mice. The expression levels of the inflammatory markers and tumor markers in the model group were significantly increased, while the expression of pulmonary fibrosis marker proteins in the positive control group was down regulated, and the expression of tumor proliferation marker proteins in the cyclophosphamide group was downregulated. The expression levels of pulmonary fibrosis marker proteins and tumor proliferation in the rhTβ4 treatment group were both significantly downregulated. Based on the different levels of protein expression, an immunohistochemical score was calculated. The immunohistochemical staining index score also reduced proportionally to the concentration of rhTβ4, indicating that rhTβ4 has a good therapeutic effect.

### 2.3. Exogenous rhTβ4 Inhibits the Proliferation and Migration of Lung Fibroblasts and Lung Cancer Cells

To further verify the effect of rhTβ4 on suppressing IPF-LC, we evaluated the effects of rhTβ4 on the proliferation and migration of lung cancer and lung fibroblasts. We created the proliferation curve of A549 cells within 72 h. The results showed that different concentrations of rhTβ4 significantly inhibited the proliferation of A549 cells at 72 h (Figure 4A). In addition, the expression level of Ki-67 protein, which marks tumor cell proliferation, was decreased in a dose-dependent manner by rhTβ4 at concentrations from 200~1000 μg/mL (Figure 4B).

A wound healing assay was used to evaluate the effect of rhTβ4 on cell migration. The results showed that the migration distance of A549 cells was inhibited by rhTβ4 in a dose-dependent manner (Figure 4C,D). The proliferation and migration of pulmonary fibroblasts can promote the progression of pulmonary fibrosis. Many studies have reported that TGF-β1 can induce the proliferation of pulmonary fibroblasts. Our results showed that rhTβ4 significantly inhibited the proliferation and migration of Mlg cells induced by TGF-β1 in a dose-dependent manner (Figure 4E,G,H). We also detected the expression of α-SMA in Mlg cells, a marker of activation of pulmonary fibroblasts to myofibroblasts. The expression of α-SMA was decreased after treatment with rhTβ4 for 24 h (Figure 4F).

### 2.4. Inhibitory Effect of Exogenous rhTβ4 on the JAK2/STAT3 Signaling Pathway In Vitro and IPF-LC Animal Model In Vivo

To investigate the mechanism of action of rhTβ4 in IPF-LC, we evaluated the effects of rhTβ4 on JAK2/STAT3, which is well-known to be activated in IPF. Firstly, we performed STAT3 report assays in STAT3 stably transfected HEK-293 cells. The results showed that rhTβ4 had an inhibitory effect on the IL-6 induced STAT3 activities which were more significant at 1000, 2000, and 4000 μg/mL (Figure 5A). An MTT assay was used to evaluate the cell toxicity of rhTβ4 on HEK-293 cells. The results showed that rhTβ4 had no or a weak toxicity on HEK-293 cells (Figure 5B).

Next, Western blotting of protein expression was carried out to test the JAK2/STAT3 pathway in A549 and Mlg cells (Figure 5C,D). The results showed that rhTβ4 inhibited the phosphorylation of JAK2/STAT3 proteins in lung cancer cells (A549) and lung fibroblasts (Mlg). Furthermore, IHC staining results revealed that, compared with the model group, the nintedanib- and cyclophosphamide-treated groups showed reduced phosphorylation of JAK2/STAT3 protein in the IPF-LC mouse model, but the decreasing trend was more significant in the high-dose rhTβ4 groups (Figure 5E,F).

## 3. Discussion

Idiopathic pulmonary fibrosis is a chronic and progressive interstitial lung disease of unknown cause, and there is no effective treatment at present. Different complications may occur during the disease progression of IPF patients, such as emphysema, lung cancer, gastroesophageal reflux, pulmonary hypertension, and obstructive sleep apnea [21]. Among them, lung cancer is a common complication of IPF. Compared with normal people, 22% of IPF patients can develop primary lung cancer, and the risk of lung cancer in IPF patients is increased by nearly five times [3].

There are similar risk factors and pathogenic mechanisms between IPF and lung cancer. IPF is closely related to smoking, and smoking is also a major risk factor for lung cancer. The vast majority of patients with IPF-LC have a history of smoking. IPF and LC are both serious lung diseases; they share a common biological pathway, and their pathogenesis is similar. There is abnormal activation of some of the same signaling pathways in both IPF and LC, including the TGF-β signaling pathway, the PD-L1/PD-1 signaling pathway [22], the PI3K/AKT signaling pathway and the Wnt/β-catenin signaling pathway [10].

At present, there is a lack of consensus on the treatment of IPF-LC. Surgery, radiotherapy and chemotherapy may increase the risk of AE-IPF and increase the mortality of patients. With the in-depth study of the copathogenic mechanism of IPF and LC, the development of anticancer and antifibrosis multi-effect drugs acting on the common mechanism of IPF and LC and combinations of antifibrosis and anticancer drugs will show certain therapeutic benefits in the treatment of IPF-LC patients.

It has been reported that in animal and clinical trials, exogenous Tβ4 has therapeutic effects on a variety of diseases, including xerophthalmia, myocardial infarction, myocardial ischemia—reperfusion injury, hepatorenal fibrosis, ulcerative colitis, colon cancer and skin trauma. Enrico Conte and his colleagues confirmed the therapeutic and protective effects of exogenous Tβ4 on bleomycin-induced pulmonary fibrosis in mice [23,24]. Although Tian et al. found that the expression of Tβ4 was increased in human and mouse pulmonary fibrosis tissues, exogenous Tβ4 improved LPS-induced pulmonary injury and pulmonary fibrosis in mice by inhibiting inflammation and oxidative stress [16].

However, at present, there are few studies on exogenous Tβ4 applied to treat pulmonary fibrosis, and its anti-pulmonary fibrosis mechanism is not clear. There are a few studies on Tβ4 in lung cancer, and although a small amount of clinical data shows that Tβ4 in lung cancer tissues is higher than in normal tissues, the effect of exogenous Tβ4 on lung cancer has not been reported. In our study, we created an animal model of IPF combined with LC. As far as we know, this is the first animal model where IPF and LC coexist in the same test subjects. We confirmed the therapeutic effect of exogenous rhTβ4 on the IPF and on the LC in our animal model. Exogenous rhTβ4 can significantly improve the impairment of the lung function and the severity of the damage of the alveolar structure by the pulmonary fibrosis as well as the growth of the lung tumor. In addition, rhTβ4 can also effectively inhibit the proliferation and migration of lung cancer cells and pulmonary fibroblast cells in vitro. However, data from the animal model in our study confirmed that the exogenous thymosin beta 4 had effects on IPF and lung cancer as two separate diseases and the treatment did not exacerbate IPF. In the follow-up study, an in-situ lung cancer model should be established to further evaluate the anti-tumor effect of exogenous thymosin beta 4 in in-situ lung cancer model and IPF-LC model.

An abnormal inflammatory reaction is one of the common pathogenic factors of IPF and LC. A large number of studies have confirmed that inflammation is closely related to the occurrence and development of cancer, and tumor tissues are often accompanied by a large amount of inflammatory cell infiltration. The constitutive activation of the JAK/STAT signaling pathway, which is an important pathway related to inflammation, is closely related to the occurrence, development, metastasis and drug resistance of lung cancer [25]. In IPF, fibrosis-related macrophages can produce fibrogenic cytokines that contribute to the inflammatory response and promote the progression of fibrosis. The activation of the JAK2/STAT3 signal pathway by various cytokines, growth factors including IL-6 is related to the pathogenesis of IPF. Ceclilia M. et al. reported that the combination of IL-6 to IL-6R and gp130 can activate JAK/STAT-mediated signal transduction to promote pulmonary inflammatory infiltration and pulmonary fibrosis in mice [26]. In this study, we found that exogenous rhTβ4 had an inhibitory effect on the IL-6 induced JAK2/STAT3 activity with a reporter assay in STAT3 stably transfected 293 cells. In addition, the results of Western blots showed that phosphorylation of JAK2 and STAT3 protein was inhibited by the treatment of exogenous rhTβ4 in both a lung cancer cell line and a lung fibroblasts cell line in vitro. Furthermore, the results of immunohistochemistry showed that exogenous rhTβ4 markedly inhibited the phosphorylation of the JAK2/STAT3 signaling pathway on the lung tissues from IPF-LC mice model in vivo. IPF is an independent risk factor for LC and chronic inflammation is one of the main causes of IPF-LC.

Although it is known that the exogenous rhTβ4 has antifibrotic effect, its targets and working mechanism are not clear. It is also unclear if exogenous rhTβ4 has antitumor effects on lung cancers, especially those occurring on the background of pulmonary fibrosis.

In this study, we found that exogenous rhTβ4 could inhibit proliferation of lung cancer cell line and delay the progression of LC in our IPF-LC model. Our studies also suggested the inhibition of IL-6/JAK2/STAT3 signal transduction by the exogenous rhTβ4 maybe part of reason for its inhibitory effect on IPF and lung cancer. To confirm that inhibition of the phosphorylation of JAK2/STAT3 by the exogenous thymosin was the responsible mechanism, overexpression and/or knock down of these proteins should be considered in future experiments.

## 4. Materials and Methods

### 4.1. Cell Culture

Mouse lung fibroblast cells (Mlg, ATCC, No. CCL-206), HEK-293 cells (KeyGEN BioTECH, China, cat: KG345) and STAT3-luc reporter stably transfected HEK-293 cells (established in our own lab) were grown in DMEM (KeyGEN BioTECH, Nanjing, China) supplemented with 10% fetal bovine serum (FBS, ExCell Bio, Shanghai, China). A549 cells (KeyGEN BioTECH, China, cat: KG007) were grown in RPMI 1640 (KeyGEN Biotech, China) with 10% FBS. LLC-luc cells (FuHeng BioLogy, Shanghai, China) were grown in RPMI 1640. The cells were maintained in a 37 °C atmosphere of 95% humidified air and 5% CO_2_. Mlg cells was used to evaluate the inhibitory effect of Tβ4 on proliferation and migration of lung fibroblasts in vitro, A549 cells were used to evaluate the inhibitory effect of rhTβ4 on proliferation and migration of lung cancer cells in vitro, HEK-293 cells were used to evaluate the effect of Tβ4 on the activity of HEK-293 cells, STAT3-luc reporter stably transfected HEK-293 cells were used to evaluate the inhibitory effect of Tβ4 on the activation of JAK2/STAT3 signaling pathway, LLC-luc cells were used to establish an in-situ animal model of lung cancer.

### 4.2. Animals

Male C57BL/6 mice (6–8 wk, 22–24 g) were purchased from Charles River Laboratories (Beijing, China). The mice were kept at a controlled temperature (22–26 °C), humidity (60 ± 2%), and under a 12 h light–dark cycle, allowing free access to food and water. All animal care and testing procedures were conducted in accordance with the standards approved by the Institutional Animal Care and Use Committee (IACUC) of Nankai University (license number: SYXK 2014-0003).

### 4.3. IPF-LC Animal Model Establishment

On Day 0, mice were grouped according to the experimental needs. After weighing and recording the weight of the mouse, it was fixed on the operating table, the neck was sterilized with 70% alcohol, an opening approximately 1 cm long was vertically cut in the neck of the mouse with a scalpel, and the tissue was separated with surgical forceps to expose the trachea. The trachea was punctured through the tracheal cartilage ring space toward the heart end, and the model group was slowly injected with a volume of bleomycin (Nippon Kayaku, Tokyo, Japan, 970592) at a dose of 2 U/kg that was suitable for its body weight. The animals were immediately held upright and then rotated left and right to evenly distribute the liquid in their lungs. In the sham-operated group, an equal amount of normal saline was injected into the trachea of the mice.

On Day 3, the injection site of the mice was disinfected with 75% alcohol, the cell suspension was drawn into a sterile syringe, and the skin was cut 1 cm below the left posterior axillary line. The subcutaneous muscle was mechanically stripped until the thoracic cavity wall was exposed and the pink lungs undulating with breathing were visible to the naked eye. A 3 mm needle was inserted vertically between the third and fourth costal arches and pierced into the lung, and 0.05 mL of 2 × 10^6^/mL LLC-luc mouse lung cancer cell suspension was injected, rotating the needle after the injection.

The animal experiment was divided into 7 groups, with 6 mice in each group. The specific groups were as follows: (1) saline; (2) bleomycin + LLC; (3) bleomycin + LLC + positive control drug (nintedanib, Beijing HWRK Chem co., Beijing, China, 656247-17-5); (4) bleomycin + LLC + positive control drug (cyclophosphamide, MedChemExpress, 50-18-0); (5) bleomycin + LLC + rhTβ4 low dose (2 μg); (6) bleomycin + LLC + rhTβ4 medium dose (20 μg); and (7) bleomycin+ LLC + rhTβ4 high dose (200 μg). The nintedanib-positive control drug group was given daily intragastric administration for 14 consecutive days from the 7th day after modeling, while the cyclophosphamide-positive control drug group was administered with intraperitoneal injection every other day from the 7th day after modeling to the 21st day. The rhTβ4 stock solution stored in PBS buffer (pH 7.4) was diluted to 20 μg/mL, 200 μg/mL and 2000 μg/mL with normal saline, respectively, and each mouse was given 100 μL of rhTβ4 solution by atomization inhalation. From the 7th day after modeling, rhTβ4 was nasally administered for 14 consecutive days after replacement with anesthesia, and the blank control group and model group were given the same amount of drug solvent.

### 4.4. Hydroxyproline Assay

The mice were sacrificed on the 21st day after BLM modeling. The right lungs of the mice were separated, put into a 5 mL ampoule, dried in an oven at 120 °C, and adjusted to pH 6.5–8.0 after hydrochloric acid hydrolysis. The residue was filtered, and PBS was added to adjust the total volume. From 10 mL, a 50 μL sample was taken, and 350 μL deionized water and 200 μL chloramine T solution was added and incubated at room temperature for 20 min, 200 μL perchloric acid was added and incubated at room temperature for 5 min, and 200 μL p-dimethylaminobenzaldehyde was added and incubated at 65 °C for 20 min. Then, 200 μL was placed in a 96-well plate to measure the absorbance at 570 nm of the sample, a standard curve was drawn using the standard substance absorbance, and the hydroxyproline concentration Cs of the measured sample was calculated according to the formula obtained from the standard curve. We converted this to the amount of hydroxyproline contained in the whole right lung according to the following formula: W = Cs × 8 (dilution factor of the measured sample) × 10 (total sample volume).

### 4.5. Evaluation of Pulmonary Function

On Day 21 after BLM modeling, the mice were anesthetized with sodium pentobarbital (50 mg/kg, Shanghai Civi Chemical Technology Co., Ltd., Shanghai, China, 57-33-0) and fixed on the operating table in a supine position. The neck fur was shaved, the trachea was exposed, and the trachea was bluntly separated. An incision was made near the head of the trachea and a tracheal joint of an intubation tube was inserted into the trachea and fixed with cotton thread. The mouse was transferred to a physiography platform, and a ventilator and the tracheal joint were connected (Peking BioLab Tech Co., Ltd., Beijing, China, Anires2005). The forced vital capacity of the mouse was recorded, and 0.1 s of force changes in pulmonary function indices, such as expiratory volume and pulmonary dynamic compliance, were applied.

### 4.6. Hematoxylin-Eosin Staining

The lung tissue sections of the mice were dewaxed and subjected to HE staining. First, they were stained with hematoxylin staining solution (Solarbio, Beijing, China, G1140) for 3 min, placed in pure water until they returned to blue, infiltrated with 75% ethanol for 1 min and then 95% ethanol for 5 min, and then stained with eosin staining solution (Solarbio, G1100) for 2.5 min. They were dehydrated using 90% ethanol and xylene. After dyeing, neutral resin was used to seal the slide, the resin was allowed to dry, and images were taken under a microscope (Nikon, Tokyo, Japan, 594200).

### 4.7. Masson Staining

Lung tissue sections were dewaxed with graded alcohol, followed by Masson’s trichrome staining kit (Solarbio, G1340). Item No. (G1340-7× 50 mL) neutral gum was used to seal the slides after staining, and the distribution of collagen fibers and muscle fibers was observed under a microscope after drying.

### 4.8. In Vivo Imaging of the Animals

On the 7th, 14th, and 21st days of in-situ tumor-bearing, mice in each group were injected with 120 µL fluorescein potassium salt solution (Beyotime, Shanghai, China, ST196), and after administering anesthesia through an atomizer, the fluorescence of the lung tumors in mice was acquired using a small animal in vivo imager (PerkinElmer, Albany, NY, USA, IVIS Spectrum). The expression situation, the size of the fluorescence area and the intensity of fluorescence were recorded.

### 4.9. Western Blot

After sampling on the 21st day, the smallest lobe of the right lung of the mice was homogenized and lysed to extract tissue proteins, and 40 ng of each histone was loaded into the sample. A549 cells were treated with rhTβ4 (10, 100, 200, 500, 1000 μg/mL), and Mlg cells were treated with rhTβ4 (0, 0.1, 1, 10, 100, 1000 μg/mL) for 24 h. RIPA buffer was used to lyse the cells, and the protein concentrations were determined using a BCA kit (Solarbio). Equal amounts of protein samples were separated on 8–12% gels and transferred to PVDF membranes (Merck, Branchburg, NJ, USA, ISEQ00010). The membranes were blocked in 5% nonfat milk for 1 h and incubated with primary antibodies (1:1000) overnight at 4 °C and secondary antibodies (1:10,000) for 2 h at room temperature. Then, immunoreactions were detected with ECL reagent (Affinity, San Francisco, CA, USA). Quantification of the band intensities was analyzed with ImageJ. The antibodies used included anti-Col-I (Cell Signaling Technology, Boston, MA, USA, cat.72026S), anti-α-SMA (Affinity, cat. AF1032), anti-fibronectin (Affinity, cat. AF5335), anti-Ki-67 (Affinity, cat. AF0198), anti-tubulin (Affinity, cat. AF7011), anti-p-65 (Affinity, cat. AF5006), anti-histone H3 (Affinity, cat. AF0863), anti-JAK2 (Affinity, cat. AF6022), anti-p-JAK2 (Affinity, cat. AF3024), anti-STAT3 (Affinity, cat. AF6394), anti-p-STAT3 (Affinity, cat. AF3293), and anti-GAPDH (Affinity, cat. AF7021).

### 4.10. Wound Healing Assay

A549 cells were maintained in a 24-well plate, and a sterile pipette tip was used to wound the cells once they reached confluence. After 0 h and 48 h, the cell migration distance was captured via a microscope (Olympus, Tokyo, Japan, CKX53). Mlg cells were maintained in a 24-well plate, and a sterile pipette tip was used to scratch a wound once the cells reached confluence. After 0 h, 12 h and 24 h, the cell migration distance was captured via a microscope (Olympus, CKX53). Cell migration was calculated as the percentage of wound closure.

### 4.11. Immunohistochemistry

Cancer tissue sections (5 µm thick) were deparaffinized. Antigen retrieval was applied with sodium citrate buffer. Then, the sections were incubated with anti-phospho-STAT3 (Affinity, AF7300, 1:200) and anti-phospho-JAK2 antibodies (Affinity, AF8158, 1:200) overnight at 4 °C. Following the manufacturer’s instructions, the tissue sections were incubated with biotin-labeled secondary antibodies (MAIXIN. BIO, KIT-9720, Fuzhou, China) for 10 min at room temperature. Signals were detected by diaminobenzidine tetrahydrochloride (DAB). Images of the tissue slices were visualized using a digital slide scanning system (3D Histech/Pannoramic MIDI, Jinan Danjier Electronics Co., Ltd., Jinan, China).

### 4.12. Cell Viability Analysis

Cell viability was determined using 3-(4,5-dimethylthiazol-2-yl)-2,5-diphenyltetrazolium bromide (MTT, Solarbio, 298-93-1). In brief, Mlg and HEK-293 cells were plated in 96-well plates (zero adjustment holes were set), and then different concentrations of rhΤβ4 were added after the cell density reached approximately 30~40%. After that, the 96-well plates were transferred to a constant temperature incubator and incubated for 24 h according to the experimental requirements. A549 cells were plated in 96-well plates (zero adjustment holes were set), and then different concentrations of rhΤβ4 were added after the cell density reached approximately 30~40%. After that, the 96-well plates were transferred to a constant temperature incubator and incubated for 24 h, 48 h and 72 h according to the experimental requirements. Then, 20 µL of MTT (5 mg/mL) was added to each well and incubated for 4 h. The old culture medium was discarded, and 200 µL DMSO (Solarbio, D8371) was loaded into each well. Finally, a microplate reader (Thermo Scientific, Waltham, MA, USA, Multiskan FC) was used to detect the absorbance at a wavelength of 570 nm.

### 4.13. Luciferase Assay

For JAK/STAT3 luciferase detection, STAT3 reporter stably transfected HEK-293 cells were plated in a 96-well plate and incubated with IL-6 (70 ng/mL, Peprotech, NJ, USA) and different concentrations of rhΤβ4 for 24 h. Then, the cells were lysed, and the luciferase value was evaluated by a luciferase reporter assay (Promega, Madison, WI, USA, E5351).

### 4.14. Statistical Analysis

Data processing and statistical analysis in this study conformed to pharmacological experimental design and analysis standards. Data are presented as the mean ± standard deviation (SD), and statistical analysis was performed using GraphPad Prism 7.0 software. After testing the data for normality and homogeneity of variance, Student’s t-test, ANOVA, and statistical analysis of multivariate data were used. The experiments were repeated three or more times, and *p* < 0.05 was considered statistically significant. * *p* < 0.05, ** *p* < 0.01, *** *p* < 0.001, NS: not significant.

## 5. Conclusions

In this study, we evaluated an animal model of pulmonary fibrosis complicated with LC. In vivo and in vitro studies with the model showed that exogenous rhTβ4 alleviated the impairment of lung function and severity of damage of the alveolar structure by the pulmonary fibrosis and inhibited the proliferation of LC tumor growth, which possibly associated with Tβ4 inhibitory effect on the JAK2/STAT3 signaling pathway. Exogenous rhTβ4 can be potentially used for the treatment of IPF and LC.

## Figures and Tables

**Figure 1 ijms-24-03818-f001:**
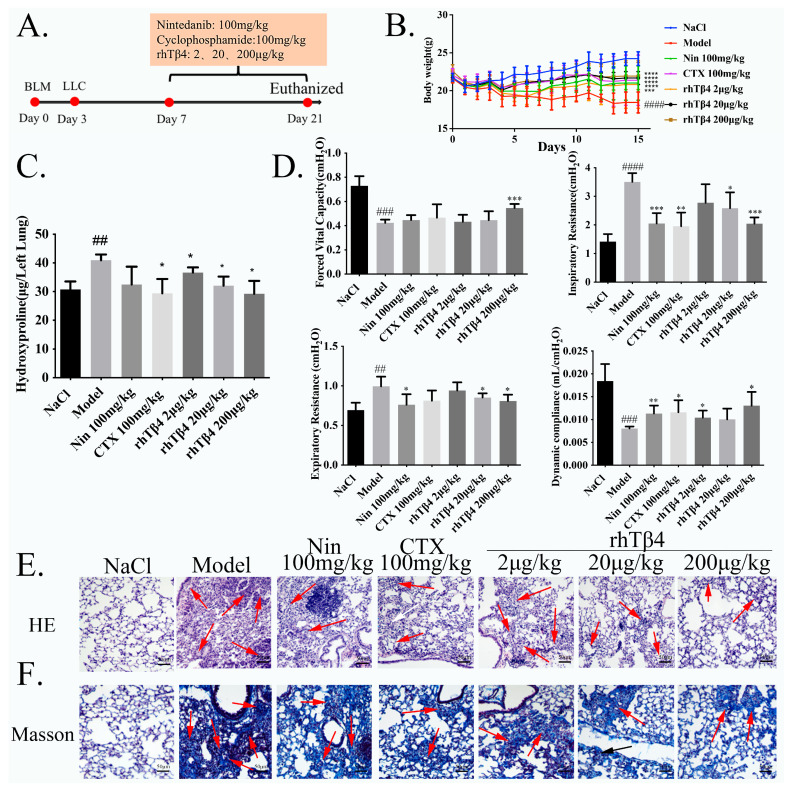
Exogenous rhTβ4 alleviates pulmonary fibrosis in mice. Effect of rhTβ4 on pulmonary fibrosis and improvements in pulmonary function in BLM-treated mice. (**A**) Dosing regimen in the BLM-induced pulmonary fibrosis model. (**B**) Mouse weight in the BLM-LLC model. (**C**) The content of hydroxyproline in the lung tissue of each group. (**D**) Lung function parameters, including forced vital capacity (FVC), dynamic compliance (Cydn), inspiratory resistance (Ri) and expiratory resistance (Re). (**E**) Lung tissue sections were stained with hematoxylin-eosin (HE) (fibrous focies were marked with red arrows), 20x, Scale bar = 50 μm. (**F**) Lung tissue sections were stained with Masson trichrome and Sirius red (fibrous focies were marked with red arrows), 20x, Scale bar = 50 μm. Data are shown as the mean ± SD. # represents the difference between the NaCl- and BLM-treated groups, ## *p* < 0.01, ### *p* < 0.001, #### *p* < 0.0001. * represent the difference between the BLM-treated and treatment groups, * *p* < 0.05, ** *p* < 0.01, *** *p* < 0.001, **** *p* < 0.0001.

**Figure 2 ijms-24-03818-f002:**
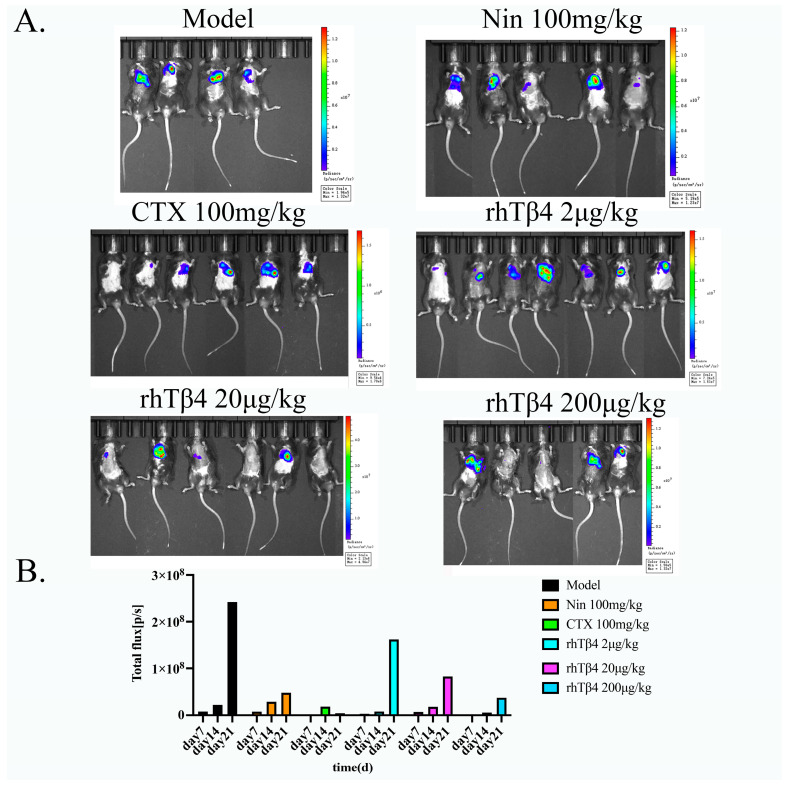
Exogenous rhTβ4 inhibits the growth of lung cancer in mice. (**A**) In vivo imaging of mice in each group. (**B**) Total fluorescence intensity of the mice in each group on the 7th, 14th and 21st days.

**Figure 3 ijms-24-03818-f003:**
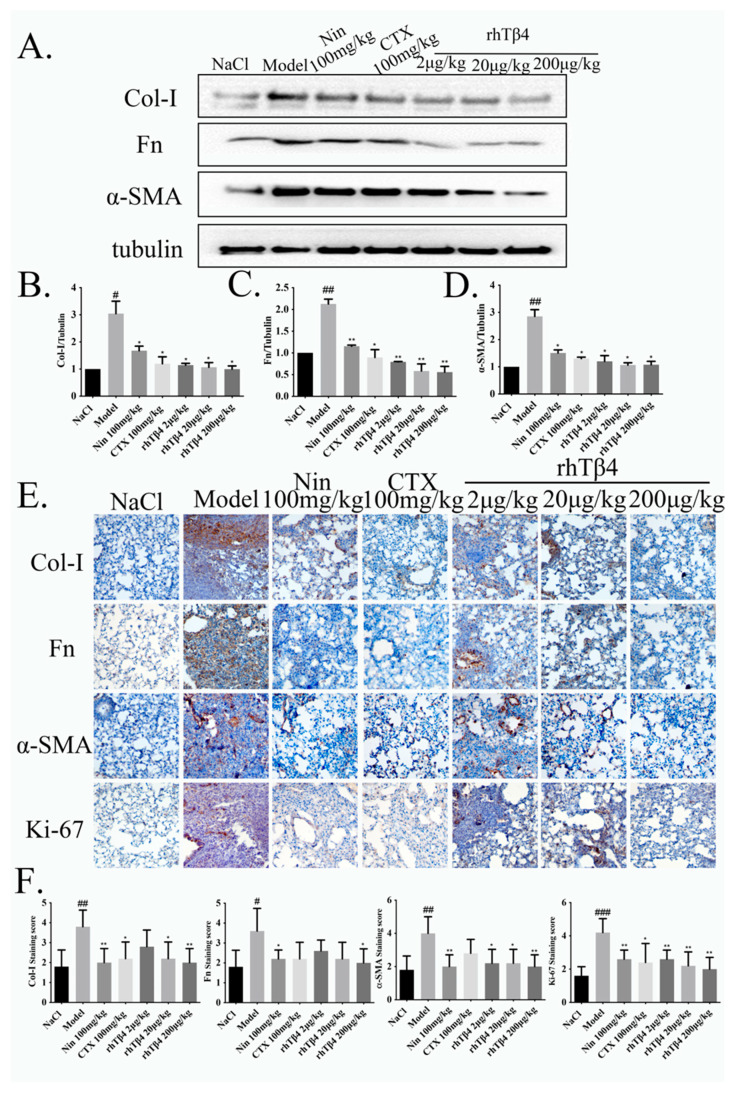
Exogenous rhTβ4 inhibits ECM accumulation and tumor proliferation in the IPF-LC model mice. (**A**–**D**) The expression levels of collagen Ⅰ, fibronectin and α-SMA in the lung tissues of mice were detected by Western blotting. (**E**,**F**) Immunohistochemical staining analysis of collagen I, fibronectin, α-SMA and Ki-67, 20x, Scale bar = 50 μm. Data are shown as the mean ± SD. # represents the difference between the NaCl and model groups, # *p* < 0.05, ## *p* < 0.01, ### *p* < 0.001. * represents the difference between the model and treatment groups, * *p* < 0.05, ** *p* < 0.01.

**Figure 4 ijms-24-03818-f004:**
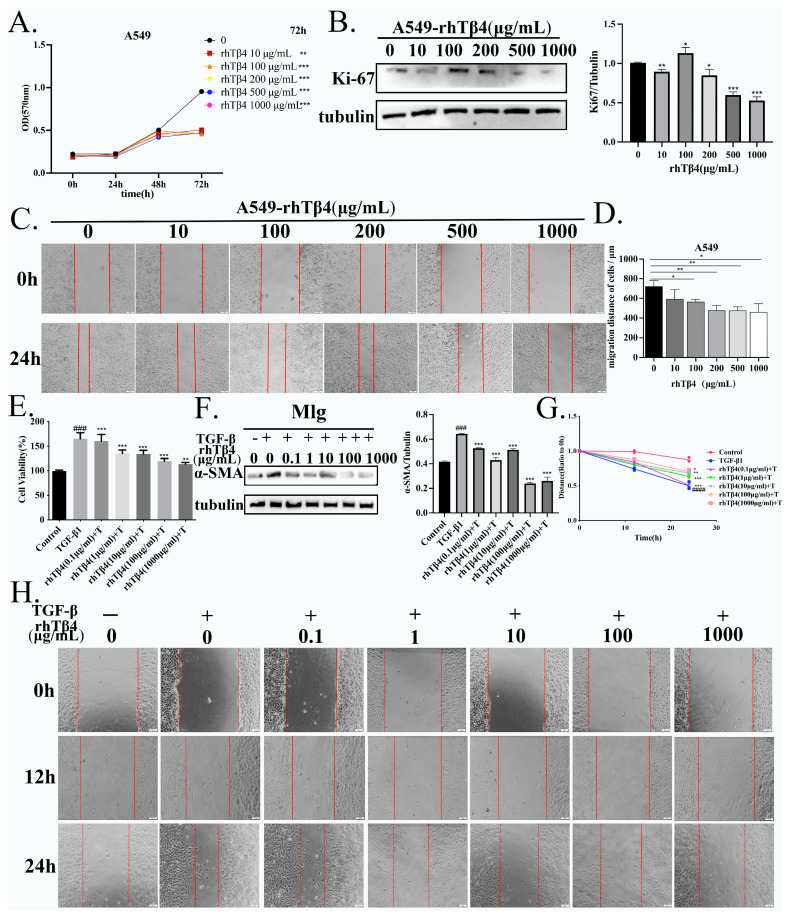
Exogenous rhTβ4 inhibits the proliferation and migration of lung fibroblasts and lung cancer cells. (**A**) The proliferation curve of A549 cells within 72 h. (**B**) The expression of Ki-67 in A549 cells treated with rhTβ4 for 24 h was detected by Western blot assay. (**C**,**D**) rhTβ4 inhibited A549 cell wound healing after 24 h. (**E**) rhTβ4 attenuated the proliferation of Mlg cells induced by TGF-β1. (**F**) Western blot analysis of α-SMA expression in Mlg cells after treatment with rhTβ4 for 24 h. (**G**,**H**) rhTβ4 suppressed the migration of Mlg cells induced by TGF-β1. ### *p* < 0.001. #### *p* < 0.0001, * *p* < 0.05, ** *p* < 0.01, *** *p* < 0.001.

**Figure 5 ijms-24-03818-f005:**
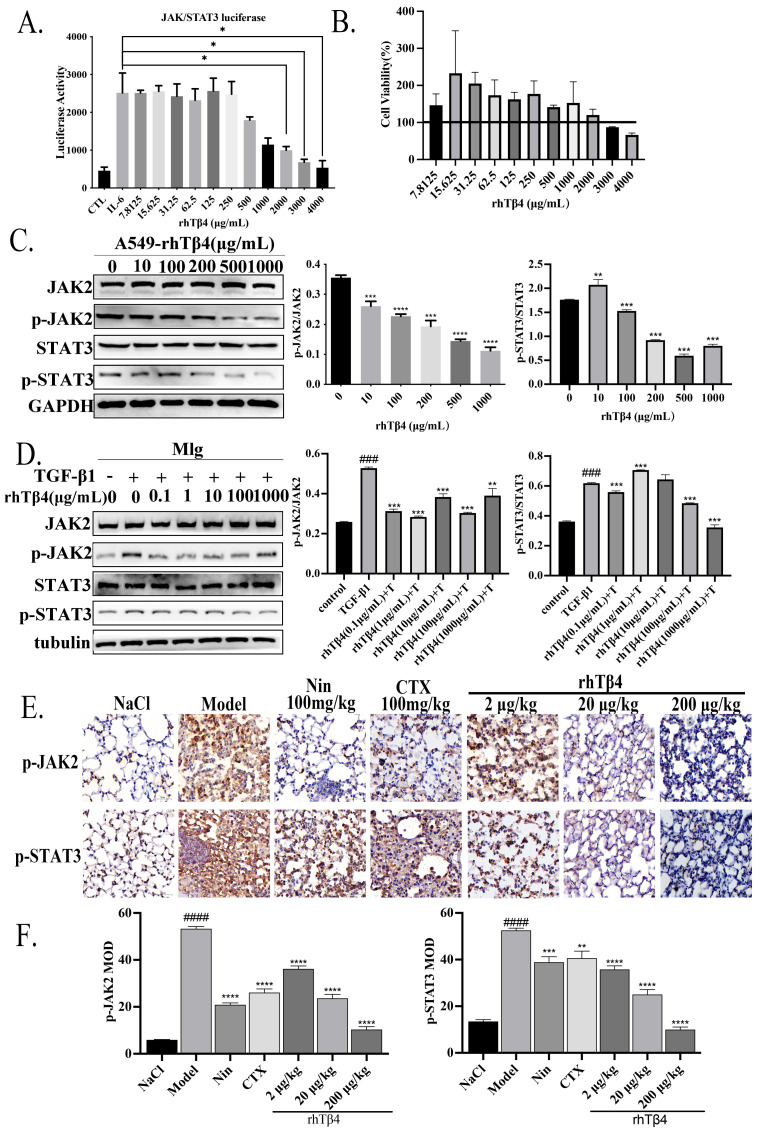
Exogenous rhTβ4 inhibits pulmonary fibrosis and lung cancer via the IL6/JAK2/STAT3 signaling pathway. (**A**) The luciferase activity in the STAT3-SIE-luc-293 cell lysates treated with IL-6 (70 ng/mL) and rhTβ4. (**B**) The viability of rhTβ4-treated HEK-293 cells. (**C**) Western blot assay of JAK2, p-JAK2, STAT3 and p-STAT3 expression in rhTβ4-treated A549 cells. (**D**) Western blot assay of JAK2, p-JAK2, STAT3 and p-STAT3 expression in rhTβ4-treated Mlg cells. (**E**,**F**) Immunohistochemical staining analysis of p-JAK2 and p-STAT3 in mice. Scale bar = 50 μm. Data are shown as the mean ± SD. ### *p* < 0.001.#### *p* < 0.0001. * *p* < 0.05, ** *p* < 0.01, *** *p* < 0.001, **** *p* < 0.0001.

## Data Availability

Data available on request from the authors.

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
