# Peer review of "Exogenous Thymosin Beta 4 Suppresses IPF-Lung Cancer in Mice: Possibly Associated with Its Inhibitory Effect on the JAK2/STAT3 Signaling Pathway"

_ijms, 2023, doi:10.3390/ijms24043818_

Round 1

Reviewer 1 Report (Previous Reviewer 1)

Authors addressed all the comments I raised and Manuscript is sufficiently  improved for the publication.

Author Response

Thanks for your comments.

Reviewer 2 Report (New Reviewer)

Overall the quality of images should be improved. Most of the IHC data can not be read from my perspective. The quality of some western blots is too poor to be published. I am concerned about the standard derivation (SD) values presented in some figures (such as Fig.5c and 5d). I can not see any SD values. Can the authors show their replicates?)  

Author Response

Reviewer 3 Report (New Reviewer)

As the author claimed, IPF and lung cancer shares many common causes from genetic background to environmental risk factors and IPF is an independent risk factor itself to the tumorigenesis of lung cancer. Majority of lung cancer post IPF is NSCLC , especially squamous cell lung cancer. In average, 30 months are needed for the lung cancer to appear on the basis of IPF. The purpose of the study is to explore the pharmaceutical effects of exogenous thymosin beta 4 on the IPF and the lung cancer in the same animal model. In the study, the author established an animal model of pulmonary fibrosis complicated with lung cancer by implanting lung cancer  cells orthotopically into the lungs of mice several days after bleomycin induction of the pulmonary fibrosis in the same mice. The advantage of the model was obvious because both the IPF and lung cancer existed in the same mice and it was convenient to test if the thymosin beta 4 had therapeutic effects on the two diseases and if there was unexpected effects happened in either disease by the treatment. The caveat here was the  implanted lung cancer cell could not be regarded as the model of lung cancer growing on the background of IPF. The implanted lung cancer cells did not grow on the background of IPF and might have histological type different from spontaneous lung cancer. Therefore, it is not appropriate to use the model to study the effects of exogenous thymosin beta 4 on the IPF-LC. That being said, data from the animal model confirmed that the exogenous thymosin beta 4 had effects on IPF and lung cancer as two separate diseases and the treatment did not exacerbate IPF.

Besides the suggestions attached to the manuscript, I have some suggestions.

1)Setup another animal study to test the effects of exogenous thymosin beta 4 on the orthotopically implanted lung cancer to observe the possible difference of the antitumor effect if there are the difference. The experiment may be helpful to answer the question about the selective antitumor effect of the agent on IPF-LC.

2)To confirm that inhibition of the phosphorylation of jak2/stat3 by the exogenous thymosin was the responsible mechanism. Overexpression and or knock down of these proteins should be considered in both vitro and in vivo experiments.

3) Simply introduce about the exogenous thymosin used in the study such as the preparation procedure and the solvent of the agent etc.

Round 2

Reviewer 2 Report (New Reviewer)

The revised manuscript has been significantly improved. Congratulations to the authors!

Author Response

Thank you for your support!

Reviewer 3 Report (New Reviewer)

Since that you will not add more mechanistic study to confirm that the inhibition of Jak/stat pathway was responsible for the effect of exogenous thymosin on IPF and lung cancer in this manuscript. you may need to consider changing the title of the study. The statement that the exogenous thymosin beta 4 suppresses IPF-lung cancer in mice via the JAK2/STAT3 signaling pathway is somewhat misleading. The title can be change to something like  "Exogenous thymosin beta 4 suppresses IPF-lung cancer in mice:  possibly associated with its inhibitory effect on the JAK2/STAT3 signaling pathway".

Author Response

Thank you for your suggestion. We have revised the title to  "Exogenous thymosin beta 4 suppresses IPF-lung cancer in mice:  possibly associated with its inhibitory effect on the JAK2/STAT3 signaling pathway".

This manuscript is a resubmission of an earlier submission. The following is a list of the peer review reports and author responses from that submission.

Round 1

Reviewer 1 Report

Rui Yu et al have identified Exogenous thymosin beta 4 improves pulmonary fibrosis and lung cancer in  IPF-lung cancer mouse model. Also the authors show that eTB4 inhibits JAK2/STAT3 signaling, thereby it improves bleomycin induced lung injury and cancer. The strength of the report is the use of several approaches including in-vivo model and in-vitro assay systems to identify the molecular mechanism eTB4 in IPF-lung cancer model. However, the following issues needs to be addressed. 

-                  - Indicate the fibrous foci in Figure 1E and 1F

-          -anti-fibrotic effect of eTB4 is known and Novelty is missing; summarize the current findings in the end of discussion

-            -rhTB4 inhibits pJAK2/STAT3 level at 200-1000ug/ml concentration but not p65 level, it is confusing. P65 must be revalidated or removed

-         - Current study reveals anti-fibrotic role of eTB4 on IPF model at its higher dose (200ug/kg), hence, it is important to validate eTB4 induced hepatotoxicity (liver injury and fibrosis) and renal fibrosis by ihc or trichrome stain.

Reviewer 2 Report

In this manuscript, Rui YU et al. analyzed the effect of exogenous thymosin beta 4 suppressing IPF-lung cancer in mice model. Furthermore, JAK2/STAT3 signaling pathway is the main signaling pathway in its tumor-suppressive and anti-fibrotic effects.

IPF-LC is s one of the important topics in lung cancer field. Overall, the experiments and analyzes are well-designed and performed. Although they have some interests, there are critical problems about the method which authors have done.

1st, in this manuscript, they use an intratracheal injection of bleomycin for lung fibrosis and injection of LLC into lung for lung cancer model. However, this model is just showing coexisting lung fibrosis and lung cancer and this model does not show any pathogenic feature of IPF-LC. Furthermore, there is just 3 days intervals between bleomycin injection and LLC injection. Therefore, this mouse model does not mimic IPF-LC at all.  

 2nd , in vitro model, they performed a functional assay of trTβ4 in lung cancer and fibroblast separately. In this assay, LC-IPF interaction which is most important in IPF-lung cancer is lost. At least, they should use the tumor-lung fibroblast coculture model. Furthermore, they use Mlg fibroblast but Mlg is totally different from cancer-associated fibroblast. Therefore they should use isolated-fibroblasts from this mouse model as cancer-associated fibroblasts.

Round 2

Reviewer 2 Report

As I mentioned in the previous review, the mice model used in the manuscript has critical concerns.